# Hepatitis B Core-Related Antigen Is Useful for Predicting Phase and Prognosis of Hepatitis B e Antigen-Positive Patients

**DOI:** 10.3390/jcm11061729

**Published:** 2022-03-21

**Authors:** Han Ah Lee, Hyun Woong Lee, Younhee Park, Hyon-Suk Kim, Yeon Seok Seo

**Affiliations:** 1Departments of Internal Medicine, Ewha Womans University College of Medicine, Seoul 07985, Korea; amelia86@naver.com; 2Department of Internal Medicine, Gangnam Severance Hospital, Yonsei University College of Medicine, Seoul 03772, Korea; 3Department of Laboratory Medicine, Severance Hospital, Yonsei University College of Medicine, Seoul 03772, Korea; younheep@yuhs.ac (Y.P.); kimhs54@yuhs.ac (H.-S.K.); 4Departments of Internal Medicine, Korea University College of Medicine, Seoul 02841, Korea

**Keywords:** immune-tolerant, hepatitis B e antigen seroconversion, hepatocellular carcinoma, hepatitis B surface antigen, antiviral

## Abstract

The role of hepatitis B core-related antigen (HBcrAg) level in defining clinical phase and predicting prognosis of chronic hepatitis B (CHB) has not been fully studied. CHB patients who had undergone liver biopsy in Korea University Medical Center were included. Patients with liver cirrhosis were excluded. The associations of HBcrAg level with CHB phase, and nucleos(t)ide analogue (NA)-induced hepatitis B e antigen (HBeAg) seroconversion were analyzed. In total, 387 patients (median follow-up of 82.4 months) were included. The CHB phases of patients were defined histologically as immune-tolerant (IT, *n* = 32, 8.3%), HBeAg-positive and immune-active (PIA, *n* = 211, 54.5%), HBeAg-negative and immune-active (*n* = 125, 32.3%), and inactive (*n* = 19, 4.9%), respectively. In HBeAg-positive patients, the mean HBV DNA levels were comparable between the two groups (*p* = 0.990). However, the mean HBsAg (7.4 log IU/mL and 6.9 log IU/mL, *p* = 0.002) and HBcrAg levels (8.2 log U/mL vs. 7.6 log U/mL, *p* < 0.001) of IT patients were significantly higher than that of PIA patients. In multivariate analysis, younger age (odds ratio [OR] 0.949, *p* = 0.025), lower alanine aminotransferase (OR 0.988, *p* = 0.002) and higher HBcrAg level (OR = 2.745 *p* = 0.022) were independent predictors of the IT phase. Of the patients in the PIA phase, 194 received NA after liver biopsy, and 61 (31.4%) had achieved HBeAg seroconversion after antiviral therapy. In Cox regression analysis, the higher HBcrAg level was the only independent predictor of the NA-induced HBeAg seroconversion (hazard ratio 1.285, *p* = 0.028). The HBcrAg level is useful for predicting clinical phase of CHB and NA-induced HBeAg seroconversion in HBeAg-positive patients.

## 1. Introduction

Chronic hepatitis B (CHB) is an important global health problem with significant morbidity and mortality [1]. It is well known that the hepatitis B virus (HBV) replication is strongly associated with disease progression and clinical outcome of CHB patients [2]. Therefore, suppressing viral replication with antiviral therapy is the major management for improving the survival and quality of life in CHB patients. However, antiviral therapy is only indicated for immune-active CHB, because no liver fibrosis and minimal necroinflammation occur in immune-tolerant (IT) and inactive (IC) CHB, indicating a low risk of disease progression [3].

The clinical phases of CHB are currently divided, according to the presence of hepatitis B e antigen (HBeAg), HBV DNA and alanine aminotransferase (ALT) levels [4]. However, clearly differentiating phases using only these markers is challenging, due to frequent inconsistency of blood tests with histologic results and patients thought to be in the “grey” zone [5]. Therefore, requirement for more reliable biomarkers in CHB has been persisted [6].

Hepatitis B core-related antigen (HBcrAg) has recently been recommended as an effective biomarker, due to its usefulness as a predictor of clinical outcomes and response related to nucleos(t)ide analogue (NA) treatment [7,8,9,10]. A recent study showed a high diagnostic performance of HBcrAg level in identification of clinical phase in HBeAg-negative CHB patients [11]. However, conflicting results have been reported regarding the role of HBcrAg for differentiating CHB phase in HBeAg-positive patients, which are not sufficient to support the general use of HBcrAg as a biomarker of clinical phase differentiation in HBeAg-positive CHB patients [7,12].

Since the CHB phases were defined by HBeAg, HBV DNA, and ALT levels in those studies, not on histological findings, interpretability of those results are limited. Therefore, this retrospective study investigated the association of HBcrAg level with the CHB phase, and NA-induced HBeAg seroconversion using only CHB patients who had undergone liver biopsy.

## 2. Materials and Methods

### 2.1. Participants

In total, 552 CHB patients who had undergone liver biopsy between 2007 and 2015 at the Korea University Medical Center were retrospectively eligible (Appendix A). The exclusion criteria were as follows: (a) age <18 years; (b) insufficient follow-up period (<6 months); (c) insufficient clinical or laboratory information; (d) histologically confirmed liver cirrhosis; (e) other causes of liver disease or co-infection with another hepatitis virus; (f) history of HCC, decompensation, organ transplant or antiviral therapy; (g) use of immunosuppressive agents and (h) other significant medical illness. The HBV genotype was not determined because >98% of Korean patients with CHB have HBV genotype C2.

### 2.2. Definition of the CHB Phase

Based on the presence of HBeAg and histological findings, patients were classified into four groups. (1) The IT group consisted of patients with positive baseline HBeAg, no fibrosis, and minimal inflammation. (2) The HBeAg-positive immune-active (PIA) group consisted of patients with positive baseline HBeAg, moderate or severe necroinflammation, and with or without fibrosis. (3) The HBeAg-negative immune-active (NIA) group was defined by the lack of HBeAg, and histology with chronic hepatitis. (4) The IC group was characterized by the HBeAg-negative, presence of serum antibodies to HBeAg and absence of significant necroinflammation with variable levels of fibrosis [4].

### 2.3. Clinical Evaluation and Follow-Up

The index date was defined as the date of the liver biopsy. During follow-up, all subjects underwent serial blood testing, including serum hepatitis B surface antigen (HBsAg), HBeAg, HBV DNA and ALT level measurements every 6 months to monitor changes from baseline. In addition, patients underwent surveillance for HCC and cirrhosis by ultrasonography and serum alpha-fetoprotein (AFP) level.

### 2.4. Liver Biopsy

Ultrasonography-guided liver biopsies were performed by two expert radiologists experienced with more than 200 liver biopsies. A transthoracic approach was routinely used with the patient in the supine position, and biopsy was performed using an 18-gauge Tru-cut needle (Speedybell™, Biopsybell, Mirandola, Italy). Each liver tissue was analyzed by three experienced pathologists. Minimum adequacy of specimen was defined as 11 or more portal tracts and a length of longer than 2 cm [13]. Knodell and Ishak scoring system were used for specimen interpretation [14,15].

### 2.5. Laboratory Assays

Serum samples collected at the index date were retrospectively studied. HBcrAg was measured by chemiluminescence immunoassay on the LUMIPULSE G12000 automated analyzer (Fujirebio, Tokyo, Japan) in the Severance Hospital, Seoul, following the manufacturer’s instructions. The range of HBcrAg quantification was 3.0–7.0 log U/mL. Samples with HBcrAg > 7.0 log U/mL were diluted and retested to measure HBcrAg level. Highly-sensitive HBsAg was measured using a two-step sandwich immunoassay method on the LUMIPULSE G1200 (Fujirebio, Tokyo, Japan). The highly sensitive HBsAg quantification range was 5.0–150,000 mIU/mL. Samples with HBsAg > 150,000 mIU/mL were diluted and retested to measure the HBsAg level.

### 2.6. Outcomes

The primary outcome was the clinical phase of CHB, and the secondary outcome was the NA-induced HBeAg seroconversion. Patients who met the reimbursement criteria for antiviral therapy in Korea were treated as follows: when ALT level was ≥2 times the upper limit of normal and HBV DNA ≥ 20,000 IU/mL in HBeAg-positive CHB patients, or HBV DNA ≥ 2000 IU/mL in HBeAg-negative CHB patients [16]. Other patients who did not meet the reimbursement criteria were monitored every 3–6 months, to detect the optimal initiation timing of antiviral therapy.

### 2.7. Statistical Analysis

Statistical analysis was performed using Statistical Package for the Social Sciences (SPSS version 25.0, Chicago, IL, USA). Data are expressed as means ± standard deviations or numbers with percentages. Differences between continuous variables were examined by Student’s *t*-test (or the Mann–Whitney test, as appropriate), and categorical variables were examined by the chi-squared test (or Fisher’s exact test, as appropriate). Associations of HBcrAg level with HBV DNA and HBsAg level were determined using Pearson’s or Spearman’s correlation coefficient, as appropriate.

The usefulness of HBcrAg to differentiate between the IT and PIA phases was evaluated by area under the receiver operating characteristic curve (AUROC), and the best cut-off HBcrAg level was derived by Youden’s index. Multivariate analyses were performed to investigate the independent predictors of clinical outcomes using the Cox proportional hazards model. Two-sided *p* values < 0.05 were considered significant.

## 3. Results

### 3.1. Patient Characteristics

A total of 387 patients followed up for a median of 82.4 (38.7–125.7) months were included. The clinical phase of each patient was defined histologically as IT (*n* = 32, 8.3%), PIA (*n* = 211, 54.5%), NIA (*n* = 125, 32.3%) and IC (*n* = 19, 4.9%). The clinical characteristics of HBeAg-positive and negative patients are presented in Appendix A. HBeAg-positive patients were significantly younger than HBeAg-negative patients (37.6 years vs. 46.0 years, *p* < 0.001). The mean HBV DNA and HBsAg levels of HBeAg-positive patients were significantly higher than those of HBeAg-negative patients (all *p* < 0.001). Finally, the mean HBcrAg level was significantly higher in HBeAg-positive patients than that of HBeAg-negative patients (7.7 log U/mL vs. 5.1 log U/mL, *p* < 0.001).

### 3.2. Hepatitis B e Antigen-Positive Patients

The clinical characteristics of the patients in the IT and PIA groups are presented in Table 1. IT patients were significantly younger than PIA patients (31.8 vs. 38.5 years, *p* = 0.004). IT patients had significantly lower ALT level (*p* < 0.001) and higher platelet count (*p* = 0.015) than PIA patients. The mean HBV DNA levels were comparable between the two groups (*p* = 0.990). However, the mean HBsAg (7.4 log IU/mL and 6.9 log IU/mL, *p* = 0.002) and HBcrAg levels (8.2 log U/mL vs. 7.6 log U/mL, *p* < 0.001) of IT patients were significantly higher than that of PIA patients (Figure 1).

### 3.3. Predictors of the Immune-Tolerant Phase

Logistic regression analyses for CHB phase were performed in HBeAg-positive patients (Table 2). Univariate analysis revealed that younger age, higher platelet count, lower ALT level, and higher HBsAg and HBcrAg levels were significantly associated with the IT phase (all *p* < 0.05). In multivariate analyses, younger age (odds ratio [OR] 0.949, *p* = 0.025), lower ALT level (OR 0.988, *p* = 0.002), and higher HBcrAg level (OR 2.745, *p* = 0.022) were independent predictors of the IT phase. The AUROC of HBcrAg to differentiate the IT and PIA phase was 0.722 (95% CI 0.625–0.818, *p* < 0.001). The best cut-off value was 8.3 log U/mL, with a sensitivity of 71.0% and specificity of 72.9% (Figure 2).

### 3.4. Predictors of the Hepatitis B e Antigen Seroconversion after Antiviral Therapy

Of the patients in the PIA phase, 194 received NA after liver biopsy, and 61 (31.4%) had achieved HBeAg seroconversion after antiviral therapy. The median time of NA therapy was 48.3 (18.4–93.7) months. Predictors for NA-induced HBeAg seroconversion were investigated, and higher HBcrAg level was the only independent predictor of the NA-induced HBeAg seroconversion (hazard ratio [HR] 1.285, *p* = 0.028) (Table 3).

### 3.5. Hepatitis B e Antigen-Negative Patients

The clinical characteristics of NIA and IC patients are presented in Table 1. IC patients had significantly higher platelet count (*p* = 0.007), and lower ALT (*p* = 0.009) and higher albumin levels (*p* < 0.001) than NIA patients.

The mean HBV DNA (3.3 log IU/mL vs. 5.5 log IU/mL, *p* < 0.001) and HBcrAg levels (4.0 log U/mL vs. 5.3 log U/mL, *p* < 0.001) of IC patients were significantly lower than those of NIA patients. However, the mean HBsAg levels were comparable between the two groups (*p* = 0.859) (Figure 1). Independent predictors for the IC phase were investigated using logistic regression analysis (Appendix A). The multivariate analysis revealed that lower HBV DNA level was the only significant predictor of IC phase (OR 0.643, *p* = 0.011).

## 4. Discussion

Although antiviral therapy is generally not indicated in IT patients, it is not easy to discriminate between the IT and IA phases. This is because of the difficulty in performing liver biopsy, even though a histologic assessment is the gold standard for defining the CHB phase. In this large retrospective study, we only recruited those who had undergone liver biopsy to precisely evaluate the usefulness of potential biomarker in CHB. Significantly different levels of HBcrAg were observed between the IT and PIA patients. The higher HBcrAg level was independent predictor for the IT phase, with the best cut-off value of 8.3 log U/mL. In HBeAg-positive patients, higher HBcrAg level was the only independent predictor of the NA-induced HBeAg seroconversion.

Our study has several strengths and clinical implications. The use of HBeAg, HBV DNA and ALT levels have shown to be insufficient for differentiation between the IT and PIA phases, emphasizing the need for novel biomarkers. In our study, the mean HBcrAg level of IT patients was significantly higher than that of PIA patients (*p* < 0.001). In addition, younger age (OR 0.949, *p* = 0.025), lower ALT level (OR 0.988, *p* = 0.002) and higher HBcrAg level (OR 2.745, *p* = 0.022) were independent predictors of the IT phase; however, HBV DNA level was not. The best cut-off value for the IT phase was 8.3 log U/mL, with a sensitivity of 71.0% and a specificity of 72.9%. These results suggest that the HBcrAg level could be a valuable marker for differentiating between the IT and PIA phases, contributing to better decision-making regarding the initiation of antiviral therapy.

Given the efficacy of currently approved antivirals, treatment should be actively administered to IA patients, due to the strong correlation between serum HBV DNA levels and the risk of HCC [2,4]. On the contrary, antiviral therapy is not recommended for IT patients, because of the low risk of disease progression [17]. Additionally, low rates of HBeAg seroconversion and HBsAg loss after treatment, the safety and high cost of long-term therapy, and insufficient evidence for preventing HCC are other concerns [18]. Therefore, determining whether the patient is in the IT or IA phase is crucial. In essence, the hallmark of IT phase is no fibrosis and minimal inflammation on histology; however, liver biopsy is difficult to perform routinely in clinical practice [19]. Therefore, the IT phase has been mostly characterized by HBeAg positivity, high HBV DNA and normal ALT levels; however, it has been controversial [4,20]. In addition, some patients have atypical laboratory results for their clinical phase, and a substantial number of patients could fall within an “grey zone” [4,5]. Furthermore, ALT level could fluctuate in IT patients for reasons other than HBV infection [21,22]. Therefore, an established definition of the IT phase has not been determined.

Based on our study results, quantitative HBcrAg level is a valid predictor for the IT phase. Consistent with our results, Seto et al. reported that the median HBcrAg level in the IT group was significantly higher than that in the IA group (8.54 and 7.92 log U/mL, *p* < 0.001) [7]. Chan et al. classified patients into six groups based on longitudinal assessment, and the median HBcrAg levels were 8.8 log U/mL, 8.5 log U/mL, and 7.6 log U/mL in the IT, PIA, and HBeAg seroconversion group, respectively [12]. In contrast to the normal ALT levels of IT patients in previous studies, the mean ALT level of IT patients in our study was 80.5 IU/L [7,12]. The histologic results of IT patients with ALT level ≥ 40 IU/L showed no fibrosis and minimal inflammation in the liver, rather, they were found to have steatosis or history of alcohol consumption. Since previous studies were conducted on patients who had not undergone liver biopsy, we believe that our study has more relevant clinical implications. ALT level may also be affected by a number of factors not associated with hepatic inflammation such as body mass index, glucose, triglyceride and total cholesterol levels. In previous studies, patients who were in the IT phase but had elevated ALT levels for reasons other than CHB may have been misclassified to the IT phase, leading to a misinterpretation of the results.

Previous studies suggested that HBsAg may be useful in understanding the natural history of CHB, reporting significant difference in HBsAg levels between IT and PIA phases [23,24]. However, HBsAg level was not significantly associated with the IT phase in this study, suggesting that the HBcrAg may be more effective for determining the IT phase. A recent meta-analysis evaluating surrogate markers of intrahepatic HBV cccDNA also indicated that HBcrAg was more strongly correlated with HBV cccDNA than HBsAg (R = 0.665 vs. R = 0.475), indicating that the HBcrAg is a more valuable biomarker that reflects natural course and treatment outcome in CHB patients [25].

In this study, higher HBcrAg level was found to be the only independent predictor for NA-induced HBeAg seroconversion (HR 1.285, *p* = 0.028), indicating the role of HBcrAg as a predictor for treatment outcome. The usefulness of the HBcrAg level for the prediction of clinical outcomes has been recently investigated [12,26,27,28,29]. Sonneveld et al. reported that HBcrAg level was lower in patients who had achieved HBeAg seroconversion than in those who had not; although this association was not identified in multivariate analysis [28]. We are also aware that a decline of HBcrAg level before NA-induced HBeAg seroconversion have been reported [12,26]. Chan et al. suggested that no viral marker could predict HBeAg seroconversion ahead; however, serial monitoring could be helpful, because the HBcrAg level in the HBeAg seroconversion group dropped significantly during follow-up [12]. Due to the conflicting results between studies, further studies with histologic confirmation and serial laboratory monitoring may provide more valuable information regarding this issue.

Our study has some limitations. First, because of its retrospective design, this study has the potential selection bias, and the sequential assessment of HBcrAg was unavailable. Therefore, it was impossible to evaluate the on-treatment predictors of HBeAg seroconversion and the kinetics of HBcrAg levels during antiviral therapy. Second, since we only enrolled patients who underwent liver biopsy in single center, the sample size was insufficient. In addition, because of retrospective design, patients with definite clinical characteristics of IT or IC phases, namely who had normal ALT levels, did not undergo liver biopsy in clinical practice, resulting in the small number of these patients in present study and elevated mean ALT levels IT and IC patients. However, in patients with histologically classified IT phase and elevated ALT levels, ALT levels gradually decreased after life-style modification, supporting reliability of these results. Third, because most Korean patients have HBV genotype C2, whether this result could be repeated in other genotypes should be validated in further studies. Lastly, although liver biopsy is the current gold standard for differentiating CHB phase, sampling error could exist. Additionally, it is hard to perform biopsy routinely in clinical practice. To the best of our knowledge, the present study is the largest study investigating the usefulness of HBcrAg in CHB with histological information, having important clinical implications.

## 5. Conclusions

In conclusion, this retrospective study of histologically assessed CHB patients shows that higher HBcrAg level during pretreatment is an independent predictor of the IT phase and HBeAg seroconversion after antiviral therapy in HBeAg-positive patients. Future prospective studies with serial assessment of HBcrAg are needed to validate our findings and to find whether HBcrAg levels can reflect a functional cure. 

## Figures and Tables

**Figure 1 jcm-11-01729-f001:**
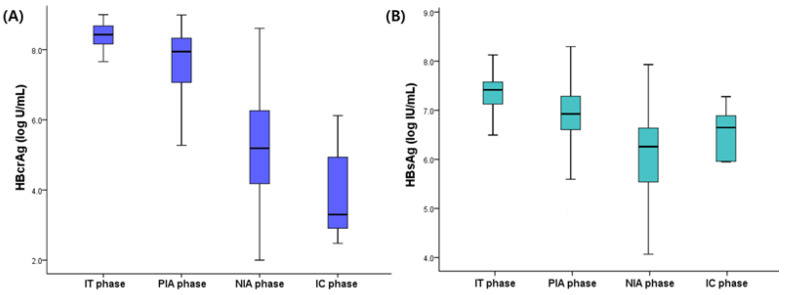
The mean HBcrAg (**A**) and HBsAg levels (**B**) according to the CHB phases. HBsAg, hepatitis B surface antigen; HBcrAg, hepatitis B core-related antigen; CHB, chronic hepatitis B; IT, immune-tolerant; PIA, HBeAg-positive and immune-active; NIA, hepatitis B e antigen-negative and immune-active; IC, inactive.

**Figure 2 jcm-11-01729-f002:**
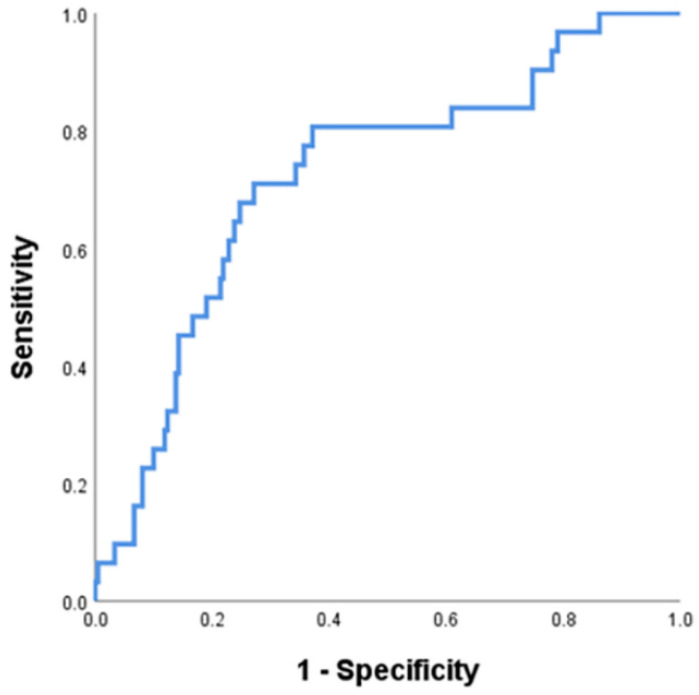
The area under the receiver operating characteristic curve analysis of HBcrAg level for predicting immune-tolerant phase. HBcrAg, hepatitis B core-related antigen.

**Table 1 jcm-11-01729-t001:** Clinical characteristics of patients according to the clinical phase of chronic hepatitis B.

Variables	IT Patients (*n* = 32, 11.9%)	PIA Patients (*n* = 211, 86.8%)	*p* Value	NIA Patients (*n* = 125, 86.8%)	IC Patients (*n* = 19, 13.2%)	*p* Value
Age, years	31.8 ± 11.0	38.5 ± 12.4	0.004	46.5 ± 10.1	43.0 ± 11.2	0.172
Male, *n* (%)	32 (100)	128 (60.7)	<0.001	86 (68.8)	14 (73.7)	0.667
Diabetes, *n* (%)	1 (3.1)	9 (4.3)	0.762	11 (8.8)	2 (10.5)	0.807
Platelet count, ×10^9^/L	210.6 ± 41.1	184.4 ± 58.5	0.015	163.0 ± 47.0	194.6 ± 49.7	0.007
PT INR	1.03 ± 0.65	1.06 ± 0.09	0.117	1.08 ± 1.00	1.06 ± 0.11	0.376
ALT, IU/L	80.5 ± 53.8	212.4 ± 242.2	<0.001	161.9 ± 259.3	77.3 ± 92.5	0.009
Bilirubin, mg/dL	0.77 ± 0.49	0.77 ± 0.48	0.987	0.97 ± 2.13	0.68 ± 0.34	0.555
Albumin, g/dL	4.07 ± 1.15	4.0 ± 0.6	0.787	4.0 ± 0.4	4.4 ± 0.4	<0.001
AFP, ng/mL	2.6 ± 1.2	17.7 ± 58.1	<0.001	11.6 ± 31.9	2.9 ± 1.3	0.253
HBV DNA, log IU/mL	7.3 ± 1.5	7.3 ± 1.1	0.990	5.5 ± 1.8	3.3 ± 1.9	<0.001
HBsAg, log IU/mL	7.4 ± 0.6	6.9 ± 0.6	0.002	5.2 ± 2.6	5.3 ± 2.8	0.859
HBcrAg, log U/mL	8.2 ± 0.7	7.6 ± 1.1	<0.001	5.3 ± 1.4	4.0 ± 1.6	<0.001

Variables are expressed as mean ± standard deviation or *n* (%). IT, immune-tolerant; PIA, HBeAg-positive and immune-active; NIA, hepatitis B e antigen-negative and immune-active; IC, inactive; INR, international normalized ratio; ALT, alanine aminotransferase; HBV, hepatitis B virus; HBsAg, hepatitis B surface antigen; HBcrAg, hepatitis B core-related antigen.

**Table 2 jcm-11-01729-t002:** Predictors for immune-tolerant phase in HBeAg-positive patients.

Variables	Rating	Univariate	Multivariate		
		*p* Value	Odds Ratio	95% CI	*p* Value
Age	years	0.005	0.949	0.906–0.993	0.025
Male sex	0 = no; 1 = yes	0.996			
Diabetes	0 = no; 1 = yes	0.763			
Platelet count	×10^9^/L	0.020	1.003	0.994–1.013	0.505
PT INR		0.117			
ALT	IU/L	0.005	0.988	0.980–0.996	0.002
Bilirubin	mg/dL	0.986			
Albumin	g/dL	0.660			
HBV DNA	log IU/mL	0.987			
HBsAg	log IU/mL	0.001	2.076	0.717–6.014	0.178
HBcrAg	log U/mL	0.003	2.745	1.157–6.514	0.022

HBeAg, hepatitis B e antigen; CI, confidence interval; INR, international normalized ratio; ALT, alanine aminotransferase; HBV, hepatitis B virus; HBsAg, hepatitis B surface antigen; HBcrAg, hepatitis B core-related antigen.

**Table 3 jcm-11-01729-t003:** Predictors for HBeAg seroconversion after antiviral therapy.

Variables	Rating	Univariate		
		Hazard Ratio	95% CI	*p* Value
Age	years	0.981	0.955–1.007	0.150
Male sex	0 = no; 1 = yes	0.736	0.440–1.233	0.245
Diabetes	0 = no; 1 = yes	0.534	0.162–1.760	0.303
Platelet count	×10^9^/L	1.000	0.996–1.004	0.904
PT INR		1.785	0.267–11.933	0.550
ALT	IU/L	1.001	0.999–1.002	0.395
Bilirubin	mg/dL	1.108	0.666–1.844	0.694
Albumin	g/dL	1.374	0.980–1.926	0.065
HBV DNA	log IU/mL	0.924	0.663–1.288	0.641
HBsAg	log IU/mL	1.056	0.574–1.942	0.861
HBcrAg	log U/mL	1.285	1.027–1.609	0.028

HBeAg, hepatitis B e antigen; CI, confidence interval; INR, international normalized ratio; ALT, alanine aminotransferase; HBV, hepatitis B virus; HBsAg, hepatitis B surface antigen; HBcrAg, hepatitis B core-related antigen.

## Data Availability

Not applicable.

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
