# Peer review of "Hepatitis B Core-Related Antigen Is Useful for Predicting Phase and Prognosis of Hepatitis B e Antigen-Positive Patients"

_jcm, 2022, doi:10.3390/jcm11061729_

Round 1
Reviewer 1 Report
In the present manuscript, Lee and colleagues retrospectively investigated the clinical role of HBcrAg for the discrimination between the different CHB phases, for the prediction of HBeAg-seroconversion in course of antiviral treatment, and for the prediction of HCC development.
Overall, the topic is interesting. The major concern regards the exclusion of patients with cirrhosis for the secondary aim of "HCC prediction". Patients with cirrhosis are those at higher risk of HCC development, and thus the more appropriate cohort for study the reliability of a biomarker for the prediction of HCC. Excluding patients with liver cirrhosis, the observed findings are at high risk of bias. Therefore, I recommend to include patients with cirrhosis, or eventually to exclude this secondary aim from the study.
Minor comments:
1) Introduction. It is not fully true that available data on HBcrAg are not sufficient to support the use of HBcrAg for CHB phases differentiation. Indeed, a recent large multicenter study showed that a singli time-point determination of HBcrAg has a high accuracy for the detection of true inactive carriers. Please amend lines 51-53, and cite the article: Aliment Pharmacol Ther. 2021 Mar;53(6):733-744.
2) Figure 1. Please add mesures of dispersion for each bar.
Reviewer 2 Report
This is a retrospective observational single-center study. Chronic hepatitis B patients requiring a liver biopsy with a least 6 months follow-up were included and divided into four groups according to HBeAg presence and histological findings. The primary outcome was the associations of HBcrAg level with the CHB phase. The secondary outcomes were the associations of HBcrAg level with nucleos(t)ide analogue (NA)-induced hepatitis B e antigen (HBeAg) seroconversion, and the development of hepatocellular carcinoma (HCC). The mean HBcrAg was higher in IT than in PIA patients at univariate and multivariate analyses. Baseline higher HBcrAg levels were associated with .a higher rate of NA-induced HBeAg seroconversion. Baseline lower HBcrAg levels associated with a higher risk of HCC development.
Histological assessment is the study's strength.
a. The authors should better explain the study population highlighting the reason for liver biopsy in all 4 subcategories. How much discrepancy is there between CHB phases between your definition and the one based on HBeAg, HBV DNA, and alanine aminotransferase levels in the population study?
b. The authors should report as a limit the baseline differences in the CHB patients groups (sex, age). The authors should try to take into account these differences in the multivariate analyses.
c. I suggest the authors revise Figure 1
d. The authors should report the average time of NA therapy. Is there a difference between patients with or without seroconversion? I suggest considering also the HbcrAg kinetics during NA treatment as shown in other HBV genotypes (like D) during NA or pegylated-interferon-α therapy.
e. The authors should declare as a generability limit the histological evaluation and the single center design (also for HBV genotype)
f. The authors should better define the population study (liver biopsy and genotype) in the conclusions. In the discussion, the authors should better support the statement "likely to be under more intense immune control"
Round 2
Reviewer 1 Report
The authors improved the manuscript as requested. I have two additional comments:
1) please provide HBcrAg values with one decimal only.
2) Figure 1A. It seems that values in the y-axis are not consistent with HBcrAg values teported in Table 1.
Author Response
1) please provide HBcrAg values with one decimal only.
Response) Thank you for your kind comment. We revised all HBcrAg values.
2) Figure 1A. It seems that values in the y-axis are not consistent with HBcrAg values teported in Table 1
Response) Thank you for your kind comment. We revised the figure 1, consistent with Table 1. Sorry for our mistake.